# Worker Protection Scenarios for General Analytical Testing Facility under Several Infection Propagation Risks: Scoping Review, Epidemiological Model and ISO 31000

**DOI:** 10.3390/ijerph191912001

**Published:** 2022-09-22

**Authors:** Jong-Myong Park, Joong-Hee Cho, Nam-Soo Jun, Ki-In Bang, Ji-Won Hong

**Affiliations:** 1Water Quality Research Institute, Waterworks Headquarters Incheon Metropolitan City, Incheon 21316, Korea; 2Incheon Research Institute of Public Health and Environment, Incheon 22320, Korea; 3Department of Hydrogen and Renewable Energy, Kyungpook National University, Daegu 41566, Korea; 4Advanced Bio-Resource Research Center, Kyungpook National University, Daegu 41566, Korea

**Keywords:** epidemiological triad model, ISO 31000 international risk management standards, non-pharmaceutical intervention (NPI), response procedures, scoping review, workplace safety

## Abstract

Infectious disease is a risk threating industrial operations and worker health. In gastrointestinal disease cases, outbreak is sporadic, and propagation is often terminated within certain populations, although cases in industrial sites are continuously reported. The ISO 31000 international standard for risk management, an epidemiological triad model, and a scoping review were the methods used to establish response procedures (scenarios) to protect workers from the risk of the propagation of a gastrointestinal disease. First, human reservoirs and transmission routes were identified as controllable risk sources based on a scoping review and the use of a triad model. Second, the possibility of fomite- or surface-mediated transmission appeared to be higher based on environmental characterization. Thus, the propagation could be suppressed using epidemiological measures categorized by reservoirs (workers) or transmission routes during a primary case occurrence. Next, using results of a matrix, a strengths–weaknesses–opportunities–threats analysis and a scoping review, the risk treatment option was determined as risk taking and sharing. According to epidemiology of gastrointestinal infections, systematic scenarios may ensure the efficacy of propagation control. Standardized procedures with practicality and applicability were established for categorized scenarios. This study converged ISO 31000 standards, an epidemiological model, and scoping review methods to construct a risk management scenario (non-pharmaceutical intervention) optimized for the unique characteristics of a specific occupational cluster.

## 1. Introduction

While global pandemics can affect industrial operations [1,2], with situations in which the propagation is limited to specific workplaces, such as in cases of gastrointestinal tract (GIT) infections, there are also risks that can hinder the normal operation of a company or industrial sites [3,4,5]. Infected workers in the test and analysis facility may not be able to properly derive the results of the experiment within a specified time, or may fail to ensure the metrological traceability of the results of the analysis. With the growth of modern industry and its dependence on the development and maintenance of diversified supply chains [6,7], if one stakeholder is affected by an infectious disease, the associated industries face expanded risks [1].

Agreement on Technical Barriers to Trade (TBT) is an international agreement that mandates each country’s standards and technical regulations to meet international standards in order to lower technology barriers to trade and revitalize free trade. For implementation of the TBT agreement, the International Organization for Standardization (ISO) provides international standards and promotes its accreditation systems based on mutual recognition agreements (MRA) [2]. Based on accreditation systems, the analytical testing service industry ensures that the quality and safety of goods are managed at a level equal to or higher than that of overseas counterparts, enabling equal and fair trade between nations [8]. In Korea, 1020 analytical testing service institutions that belong to the agriculture, aerospace, biological resources, food, medical, pharmaceutical, military and other engineering fields are accredited according to series of ISO 17025 standards.

However, with the advent of the COVID-19 pandemic, the closure of accredited analytical testing service industries resulted in expanded risks to associated industries, which led to normal trade procedures between nations being blocked. The important lesson that was learned from this experience was that industrial workplaces should prepare in advance for infectious diseases and their negative impact.

For the efficiency of the nation’s public health administration, the development of infection or propagation control guidelines may be prepared to pandemic cases [9]. However, in case of GIT infection, since an outbreak is sporadic and propagation is often terminated within certain populations in industrial sites, national guidelines specific for industrial sites are not prepared, and they tend to rely solely on the hygiene management capabilities of industrial sites (or the company). Therefore, management failure has had a serious impact on the operation of institutions. Serious cases of propagation between employees in global hotel chains (2017, typhoid fever), hypermarket chains (2018, EHEC), in the Olympic Committee (2018, norovirus) and countless under-reported cases exist in the Korea Republic. Therefore, from an economic point of view, to protect industrial groups, company or workers, appropriate response procedures for risk of GIT infection should be established proactively.

National infection control policy was developed to incorporate the characteristics of nation’s quarantine, medical delivery, and administrative systems. However, in situations wherein there is limited propagation within specific industrial sites, there may be a greater possibility of success when the risk treatment programs that are being implemented reflect the characteristics of their specific environments and management systems. The ISO 31000 international standard presents scientific strategies, methods, and procedures for risk management [10,11,12,13]. In addition, findings of studies that have been conducted have indicated the use of ISO 31000 to control chronic or subacute diseases in workplaces [14,15,16,17,18], since it provides a framework that has been optimized for the unique characteristics of workplaces. This enables the identification of risk treatment programs. In this study, GIT infection risk sources that can lead to propagation within an analytical testing facility were identified and appropriate scenarios for laboratory workers were developed using scoping review methods, the ISO 31000 and the epidemiological triad model.

## 2. Materials and Methods

### 2.1. Identification of Risk Sources (Step 1 of Risk Management)

#### 2.1.1. Epidemiological Triad Model and Scoping Review

An epidemiological triad [19,20] model was used to determine risk sources that cause (Figure 1) [10] infection propagation between lab workers, as well as methodologies to identify the etiology and guidelines for controllable risk sources that constitute the epidemiological triad model of gastro-intestinal disorder. In addition, a scoping review (Figure 2) of previous epidemiological reports was conducted (see Section 2.5) [21,22].

#### 2.1.2. Deduction of Environmental Risk Sources

Environmental characteristics are one of the major factors in the propagation of infections constituting epidemiological triad model [19,20]. The environmental characterization of general lab facility conducted aimed to develop standardized and accredited analytical testing service facilities, compliant with international standards (ISO 17025:2017) for laboratorial facility and lab workers’ performance [23]. First, the general domestic space, general office space and the unique spaces for analytical testing facility were stratified (Table 1) for precise environmental monitoring [24,25]. Second, the specific environmental risk sources (factors that caused direct or indirect transmission in the general lab facility) were identified by monitoring the laboratory workers’ occupational procedures.

### 2.2. Risk Analysis (Step 2)

For conducting risk analysis (Figure 1) [10,12], a matrix model [26] (widely used method in the public health field that enables appropriate selection of the risk treatment option (see Section 2.3) based on the quantification of the risk size) was conducted by three blinded experts who belongings to the industrial field (Ph.D. thesis on microbiology). The likelihood (L) and severity (S) were scored as 1–5 points, respectively, and risk was quantified with 1–25 points [27].

### 2.3. Deduction of Risk Treatment Option (Step 3)

To develop an optimal strategy with goals optimized for specific industrial sites, risk treatment options (direction) [12] as well as implementation and stipulation means were obtained from the results of the matrix modelling (see Section 2.2) and strengths–weaknesses–opportunities–threats (S.W.O.T.) analysis [28]. In general, risk taking, risk sharing, risk avoidance, risk-source elimination, reducing likelihood (L) or severity (S) are presented as risk treatment options according to the ISO 31000 risk management standard.

### 2.4. Establishment of Risk Treatment Plans (Step 4)

The development of the risk treatment plan designed with three stages. First, the risk scenario was constructed as a risk treatment plan, an adequate means for implementing the selected risk-treatment option (by conducting Section 2.3) ‘risk sharing’ and ‘risk taking’ and extremely higher risk severity (S) quantified by matrix model (by Section 2.2) [12]. The scenarios were prepared with regard to regulating the controllable risk sources proposed in Figure 3, which were deduced by considering the risk sources identified in Table 1, Table 2 and Table 3. Second, the risk sources related to transmission route and human reservoir (primary, further and confirmed case, close or indirect contact) were categorized and the scenario for each category was identified. Third, the applicability, practicality, and resources required for the implementation of each scenario were reviewed by three blinded reviewers using respectable workers protection guidelines for specific industrial sites under pandemic risks [24,25] as comparative indicators.

### 2.5. Scoping Review

A scoping review was conducted as (Figure 2) procedures describes [21,22]. Finally, the literature was discussed to provide a scientific basis for decisions at each stage (Step 1–4) of the risk management procedure.

## 3. Results

### 3.1. Environmental Risk Sources

The general analytical testing facility was more susceptible to direct/indirect transmission due to the unique character of its working procedure and industrial roles. Potential direct transmission (close contact) routes were identified in cases that required collaboration or co-working, such as in the sampling and pretreatment stages (where time and space are shared completely between workers) (Table 2). Even if working time was not shared, cases of potential indirect contact route through diverse fomites and surface were identified through the use of common experimental areas, facilities, and utensils (Table 2).

### 3.2. Transmission Routes as Risk Sources

In the screening stage of the scoping review (Figure 2-Step 3), it was possible to summarize constituting factors of epidemiological triad model—the infection source factors (causative agent, human reservoir), environmental factors (transmission route), and host factors (host and susceptibility) that affected occurrence and transmission (Figure 2).

Based on the fourth step of the scoping review (Figure 2-Step 4), screened and selected studies could be classified as three viewpoints, in cases A–C. Case A studies were those that are focused on host susceptibility and were conducted intensively on cases occurring in the population with specific sensitivity (immune-suppressed, hospitalized or in a care center) [29,30]. Case B were studies that focused on the etiology, physiological, and molecular characteristics of causative agents aiming at healthy populations that were segregated geographically (e.g., ethnic, rural, island, or military) [5,32,35,36,37,42,43,47,48,49,50,51,52,53,54,55,66,68,69,70,71,72]. Using this method, the laboratory isolation and identification of the causative agent, transmission routes, reproductive ratio, fatal and severity rate were revealed. Case C studies highlighted the environmental factors to show how the defined environmental characteristics affected the occurrence and transmission of diseases, enabling specific vulnerable environments (e.g., slaughterhouses, extreme poverty, near-to-sewage treatment facilities) to be targeted [39,62,73]. Among these studies, literatures that were suitable for the purpose of this study were finally selected (Figure 2-step5). The discussion and evaluation of the literature revealed several patterns. Within the healthy population, direct transmission through daily contact, such as hand contact and talking, co-working was identified, and many actual cases have also been reported continuously, which made it easy to determine the possibility of direct transmission at the analytical testing facility. However, reports did not show relative diversity of cases in terms of causative agents that caused indirect transmission in a healthy population.

Therefore, it was considered that possibility of propagation through indirect (fomite- or surface-mediated) transmission can be determined from reports that satisfy the following conditions. First, it should be a case in which the primary cases shed the causative agents (e.g., if human was the terminal host, for an infection such as a *Coxiella burnetii* infection, it should be an exclusion criterion of the scoping review). Second, verified cases of microbial states for which laboratory cultures were not possible, but according to molecular studies [38,40,44,46,51,56,57,71,74,75], activity remained on the fomites, organic/inorganic surface for a considerable period (viable but non-cultivable, VBNC) could be found [40]. The possibility of direct, indirect transmission of each causative agent based on scoping review results is discussed in the Appendix A. Note, the VBNC states of microbial agents seems to be varied by the surface of fomites materials, time elapsed after contamination, and method of conducting study [38,40,51,71,74,75]. Therefore, there may have been cases where the propagation pattern was reported differently as studies accumulated. However, it should be noted that the purpose of this study is not to academically review the propagation pattern, but to secure scenarios (response procedures) based on scientific basis obtained by the scoping review method, a supporter for decision-making.

Finally, the propagation characteristics of each agent were categorized into types in terms of the applicability of scenarios containing measures such as isolation, releasing, and close observation of workers. These measures were configured according to the categorized transmission and propagation types A–D (Table 3). Type A transmission mode included cases wherein direct transmission was confirmed by the scoping review and can also be propagated by fomite- or surface-mediated transmission. Type B were cases of amoeba infection, and person-to-person transmission was limited to sexual transmission. Due to the resistance of the cyst to environmental stress, the vegetation of cysts that remained on inorganic surfaces was identified.

In type C transmission mode, *Coxienella burnetii* or *Brucella* sp. were the causes of oral infections among complex infection routes; however, the human reservoir is not a discharging agent as an end host [64,65]. *Vibrio Vulnificus* and *V. parahemolyticus* (Table 3) showed no cases of person-to-person transmission, or the possibility was rare. In such cases, the symptomatic case did not need isolation.

Type D included the possibility of bacteria or spores shedding through feces, and this remained in fomites or surfaces due to their strong resistance. Usually, *Clostridium perfringens* or *Bacillus cereus* showed pathogenicity within a larger amount of vegetative cell or toxins that accumulated in the enteric tract. Antibodies *to C. perfringens* toxin in the wider population were confirmed. Therefore, management of potential reservoirs under shedding is an insufficient method to block propagation. In contrast, propagation should be controlled using environmental disinfection or hand hygiene practices.

As a result, it is possible to apply a categorized (type A–D) scenario for transmission pattern of causative agent based on scoping review results (Appendix A), and medical evaluation of cases should be accompanied to enable such action.

### 3.3. Human Reservoir as Risk Sources

The risk sources related to the etiology and environmental factors identified above indicate that risk sources be categorized and managed accordingly (Table 4-Part 1 and 2). Based on the scoping review, the potential human reservoir was identified (Table 4). In implementing the scenarios, it is recommended that reservoirs should be defined as part of their standard operating procedures (SOPs).

### 3.4. Risk Analysis

Likelihood, a factor that determines risk size, is close to 1; however, once it occurs, the risk may have serious impacts on industrial sites and occupation. The reason why the likelihood of occurrence was evaluated as low was that the epidemiology of GIT infection is sporadic, with the likelihood of occurrence varying greatly, depending on the social or temporal factor [76,77]. Unpredictability and possibility of inflow or transmission into the occupational group depend on the differentiable secondary reproduction ratio. In general, when the probability of occurrence is low but the intensity of the risk is high, the scenario is thought to be an appropriate risk management method.

### 3.5. Risk Treatment Option: Direction, Goal, Scenarios, and SOP

It is not possible to block disease introduced through the primary case from an external environment. Thus, the risk management direction was set as ‘risk taking’. The opportunity for propagation may be reduced by controlling deduced controllable risk sources (Figure 3) and changing the risk sources of the transmission routes and human reservoirs.

Therefore, the goal of a risk treatment plan is to delay or block propagation through categorized and stipulated epidemiological measures during primary case occurrences (Figure 3). Therefore, ‘risk sharing’ and ‘reducing likelihood (L) of occurrences’ also can be a risk treatment option. Risk sharing is to communicate between the members (workers) about the fact that infectious disease inevitably entering the organization from the external environment cannot be completely controlled. The decrease in the likelihood of occurrence means that the additional spread of infectious diseases can be controlled through the emergency response procedure developed in this study despite the inevitable inflow of infectious diseases.

Meanwhile, for most infection curves of GIT infection agents, the start points of the shedding period located behind the symptoms progress considerably [70]. Therefore, propagation can be effectively slowed down when the primary case is controlled quickly and appropriately. The risk management process means including these characteristics in the scenario [78].

SOP was selected as a means of implementing scenarios that are widely used in public health fields and in the analytical testing service industry to overcome uncertainty of risks and to enhance organizational stability and efficacy [79,80,81]. With quality controls or experimental procedures that are established and stipulated in an SOP to secure metrological traceability, the advantage is that laboratory employees may be less reluctant to document their procedures according to S.W.O.T. analysis results (Table 5). Since workers perform their work according to prescribed procedures or perform periodic documentation tasks, it may be advantageous to track the close or indirect contact history in analytical testing occupations (Table 5) as also suggested in the respectable worker protection guidelines under pandemic risk conditions [24,25]. Scenarios may also be incorporated easily into the existing working procedures, and suitable means of sharing risks between members of general analytical testing facility selected.

### 3.6. Scenarios

Scenarios were constructed according to the flow of epidemiological events that occurred over time after the primary case occurrence (Table 6 and Table 7, Figure 4). Health quarantine, observation, case isolation, and medical evaluation was defined (Table 6-Part 2) to enable appropriate management. All specific operating procedures are described (Table 6 and Table 7) and visualized by the schematic diagram (Figure 4) for easy use.

#### 3.6.1. First Step of Configuration of Procedures over Time: Primary Case Occurrence

When the primary case occurs, they should be classified and managed for each transmission route as described in the scenarios (Table 6 and Table 7). For example, since it was not known which type of transmission (A, B, C and D in Figure 4) occurred in the first case, medical evaluation was required (Table 6-Part 2, Figure 4). If the possibility of direct or indirect transmission is suspected strongly even before medical evaluation results are obtained (Table 7-Part 3, Figure 4), close and direct contacts should be identified, and close observation should be conducted.

**Table 7 ijerph-19-12001-t007:** Scenario.

Part 3. Further Case Occurrences	
**3.1**	**Health quarantine, close observation, and isolation**
(1)	Further case occurrence indicates Type A propagation, even before the medical evaluation of primary case deduction. Close and indirect contacts should be isolated from occupation. The reason for attempting indirect contact isolation is that the risk of indirect transmission increases in the case of direct transmission.
(2)	Medical evaluation and treatment must be performed in consideration of asymptomatic case that discharge causative agents in both close and indirect contacts.
(3)	If someone does not correspond to close or indirect contacts, close observation is required because of a shared general domestic environment.
**3.2**	**Record management**
(1)	Symptoms of further cases and medical evaluation results should be recorded and managed.
**Part 4. With Deduction of Medical Evaluation**	
**4.1**	**Type A**
(1)	Even if there is no further case occurrence, all close and indirect contacts must maintain isolation to block or slow down propagation and obey the procedures of (2.3)
**4.2**	**Type B**
(1)	Close and indirect contacts maintain close observation because of the possibility of indirect transmission through close and indirect contact.
(2)	Isolation should be performed whenever additional symptoms occur during close observation. Rapid propagation among close and indirect contacts can be blocked by maintaining close observation.
**4.3**	**Type C**
(1)	Primary case can be released from isolation.
(2)	Close and indirect contacts can be released from health quarantine or close observation.
**4.4**	**Type D**
(1)	Primary case might be isolated until symptom extinguished.
(2)	Close and indirect contacts might be released from health quarantine and close observation.
**Part 5. Release**	
**5.1**	**Confirmed case**
(1)	This is limited to Types A and B. Types C and D comply with the regulations of 4.3 or 4.4
(2)	Since shedding period varies by etiology and host factor, it is necessary to return to occupation after confirming that there is no discharge through medical evaluation (2.3).
(3)	If medical evaluation is not possible due to financial conditions, the case should be excluded from occupation until the longest known period of each agent’s discharging period.
**5.2**	**Health quarantine**
(1)	This is limited to Types A and B. Types C and D comply with the regulations of 4.3 or 4.4.
(2)	Close and indirect contacts without any symptoms can return to occupation after the maximum shedding periods has elapsed from the last case isolation date, and other close observations can be released.

#### 3.6.2. Second Step of Configuration of Procedures over Time: Medical Evaluation and Further Case Occurrence

Next, medical evaluation has to be conducted because additional procedures may be performed according to whether the primary case is Type A, B, C, or D (Table 7—Part 4, Figure 4). Further, to apply these scenarios, health monitoring and record management (Table 6 and Table 7—Part 2 and 3), exposure investigation, and inspections (Table 6—Part 2, Figure 4) must be conducted because of the risk severity (S) of propagation (the magnitude of the loss due to risk is very great, especially if the normal functioning of the analytical testing facility is to be maintained).

Even before the medical evaluation results are obtained, if further cases occur, propagation within the group may be confirmed based on the recorded results of the primary case (Table 6 and Table 7—Part 2 and 3, Figure 4).

#### 3.6.3. Third Step of Configuration of Procedures over Time: Categorized Scenario

Next, according to results of medical evaluation, corresponding categorized scenarios (Type A–D) can be conducted (Table 7—Part 4).

#### 3.6.4. Fourth Step of Configuration of Procedures over Time: Release

Next, the time point of release from case isolation, close observation, and health quarantine should also be established. Time to return to the workplace must be the time when the microbial shedding has stopped (Table 7-Part 5, Figure 4). However, according to the scoping review, the shedding period of a particular causative agent may continue after the end of any visible symptoms [34,62] and act as loopholes in disease control, leading to the re-occurrence of propagation. Unfortunately, the shedding period is not typical, even if taxonomic information is the same. The end of the discharging period is influenced by the host factor (sex, age, nutrition, hydration state, and medication) [32,35,70]. Therefore, ideally, after a medical evaluation, case isolation should be stopped after confirming that discharge has ceased. (Table 6-Part 2, Figure 4). In difficult instances, cases should be excluded until the longest shedding period has elapsed (Table 7-Part 5, Figure 4). It should be noted that these scenarios must be harmonized with the cleaning and disinfection of the working area.

In summary, as few scenarios have been proposed regarding how to manage such risks of infectious disease propagation in the industrial field, we aimed to provide new procedures. In general, response procedures in SOP should be suitably designed considering the overall operating factors, such as hardware (i.e., budget) or finances. However, the process of quality management must be considered. In the course of a company’s growth, it is crucial to follow the methods and procedures provided by the quality assurance system to the maximum extent, although the factors that makes up its management operation are insufficient to introduce an intact certification system. Therefore, the emergency response procedures proposed in this study seem important for protecting workers from propagating infectious diseases.

## 4. Conclusions

In conclusion, scenarios to cope with the risk of prorogation of gastrointestinal diseases suitable for the unique characters of the analytical testing facility were established by an interdisciplinary study. The scenarios identified scientific clues from the scoping review and epidemiological model. The applicability and practicality of scenario were confirmed using the ISO 31000 framework.

## Figures and Tables

**Figure 1 ijerph-19-12001-f001:**
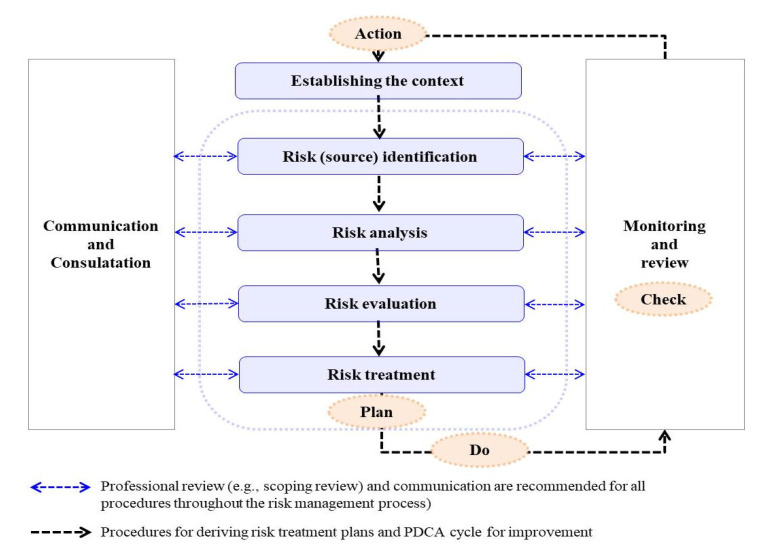
ISO 31000 risk management process.

**Figure 2 ijerph-19-12001-f002:**
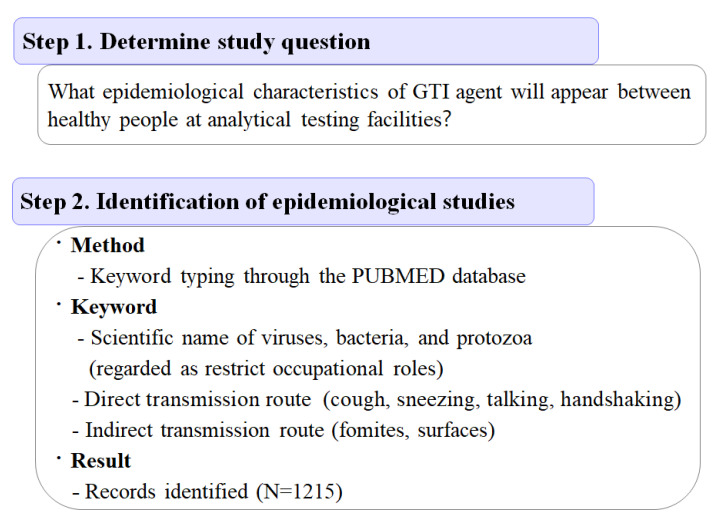
Scoping review procedure and result of each stage. In the step 2, keyword typing was used [5]. Results of data extraction (step 5) was listed in the Table 3.

**Figure 3 ijerph-19-12001-f003:**
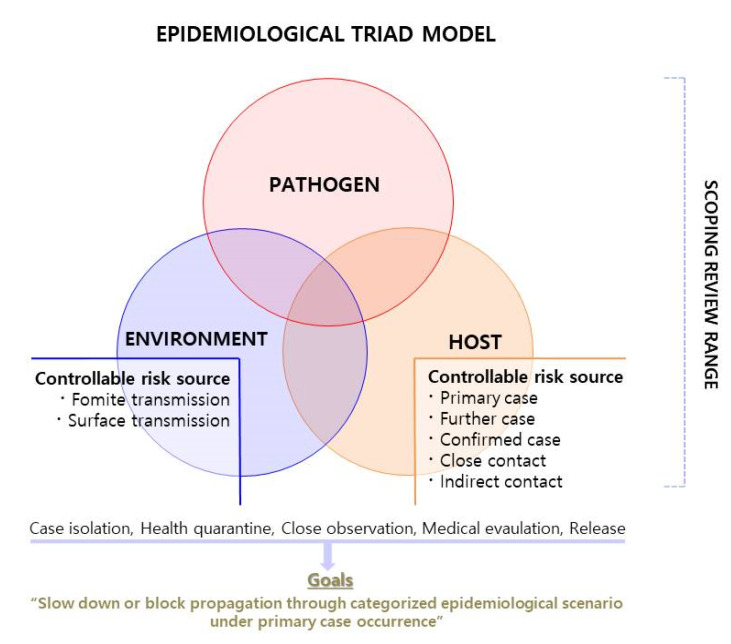
Schematic for (controllable) risk sources determination introducing the epidemiological triad model.

**Figure 4 ijerph-19-12001-f004:**
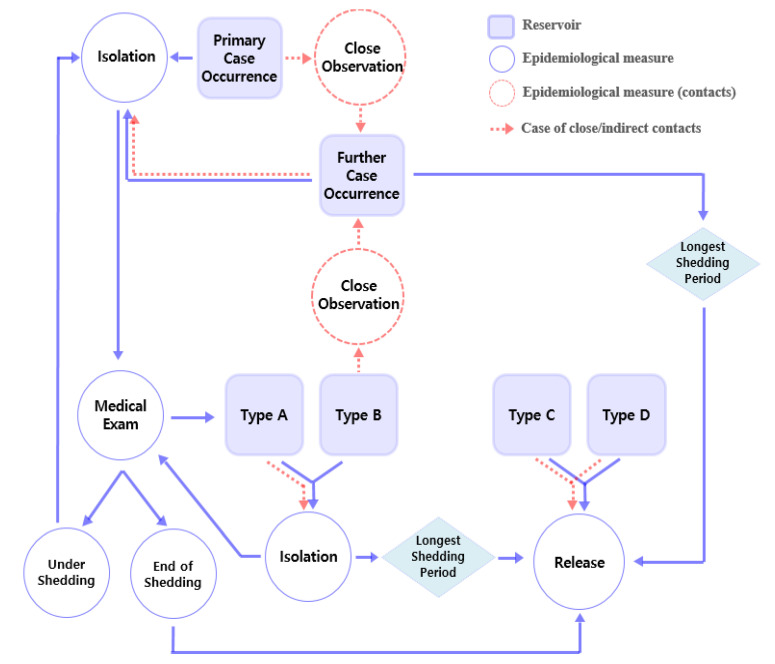
Flow diagram of response procedures (scenarios).

**Table 1 ijerph-19-12001-t001:** Stratification of the analytical testing facility environment to identify unique environmental risk sources constituting epidemiological triad model of gastro-intestinal disorder.

Stratification of Environment	Environmental Risk Sources Potentially Causing Direct or Indirect Transmission
A. General daily space	Direct contact (talking, meal, coughing)
	Indirect contact (toilet or meal facility)
B. General office space	Direct contact (meeting, hand-shaking, talking, coughing)
	Indirect contact (joint phone, handle, meeting room and door)
C. Space unique to analytical and testing facility	Direct contact (co-working, collaboration)
	Indirect contact (joint space, facility, equipment, tools)

**Table 2 ijerph-19-12001-t002:** Environmental risk sources deriving from uniqueness workspaces character of lab facility.

Transmission Route	Risk Source	
Direct transmission ^(1)^	Collaboration	Sampling (4 days, 9 people)
		Pretreatment of experimental sample (3 days, 10 people)
	Co-working	Use of communal experimental room at the same time (12 days, 11 people)
		Use of experimental desks at the same time (10 days, 11 people)
		Use of clean bench or home hood at the same time (11 days, 5 people)
Fomite or surface transmission ^(2)^	Joint space	Communal experimental room (11 count/day)
	Joint facility	Clean bench or home hood (6.5 count/day)
		Chair (8.8 count/day)
		Deionizer (9 count/day)
		Gas chromatograph (1.5 count/day)
		Inductively coupled plasma (0 count/day)
		Liquid chromatograph (1.5 count/day)
		Experimental desk (11.5 count/day)
	Joint equipment, utensils	Pipette (6.5 count/day)
		Syringe (2.8 count/day)
		Types of handles (8.5 count/day)
		Sprayer (8.8 count/day)
		Sterilizer (3.3 count/day)

^(1)^ day: as factors causing direct contact, it indicates how many times two or more collaborations have been made within 14 days of monitoring. ^(2)^ count/day: as a factor causing indirect contact, the value divided by 14 days (monitoring days) by counting how many people used the same object within the working hours (9 h) per day.

**Table 3 ijerph-19-12001-t003:** Categorization of transmission route of each microbial agents based on scoping review of previous epidemiological studies.

Categorization Type	Risk Sources
Type A. Direct, fomite- and surface-mediated transmission	Enterohemorrhagic *Escherichia coli* [29,30,31]
	*Vibrio cholerae* [32,33]
	*Salmonella typhi* [34]; *S. enteritidis* [35]; *S. typhimurium* [36,37,38]
	*Shigella* sp. [39,40,41]
	Rotavirus [42,43,44,45]
	Astrovirus [46]
	Norovirus [44,47,48,49,50,51,52,53]
	Hepatitis A virus [44,54,55,56,57]
	*Giardia intestinalis* [7,58]
Type B. Fomites or surface transmission	*Entamoeba histolytica*, *coli* [59,60,61]
Type C. Water or foodborne	*Vibro vulnificus, V. parahemolyticus* [33]
	*Campylobacter* sp. [62]
	Entero-invasive, -aggresive, -pathogenic, and -toxigenic *E. coli* [63]
	*Coxienella burnetii* [64]
	*Brucella* sp. [65]
Type D. Intoxication or atypical carrier	*Clostridium perfringens* [66]
	*Bacillus cereus* [67]
	*Staphylococcus aureus* [68,69]

**Table 4 ijerph-19-12001-t004:** Probable human reservoir and definition of terms for performing response procedures.

Part 1. Potential Reservoirs		
1.1	Primary case	First symptomatic workers with visible symptoms
1.2	Further case	Additional case under similar symptoms as the primary case
1.3	Confirmed case	Cases in which an intestinal infection or causative agent was identified in a medical evaluation
1.4	Close contacts	In cases of sharing both time and space through co-work (collaboration) with cases, e.g., sampling, pretreatment procedures, using a shared laboratory room, facility, utensils at the same time
1.5	Indirect contacts	In cases involving sharing of space and property with a time difference (if joint facilities, equipment, and utensil are shared)
**Part 2. Measures for Potential Reservoirs**		
2.1	Isolation	Exclusion from occupation of all cases or contact, according to a response procedure scenario
2.2	Close observation	All close or indirect contacts should be observed by manager with the onset of symptoms in mind, to slow down or block propagation
2.3	Health quarantine	Even if there are no symptoms, isolation is performed if there is a risk of infection after exposure to the primary or confirmed cases, to slow down or block propagation
2.4	Release	Return to occupation from isolation, close observation, health quarantine
2.5	Medical evaluation	Clinical estimation diagnosis or laboratory diagnosis by medical staff to estimate or determine the cause of symptoms.

**Table 5 ijerph-19-12001-t005:** Strengths–weaknesses–opportunities–threats analysis for preparing optimal risk treatment options for analytical testing service industry.

Internal Factor			
Strengths		Weakness	
S1	Proceduralization and standardization of work	W1	Increased possibility of close contact
S2	Acceptability of regulations and procedures to workers	W2	Increased possibility of indirect contact
S3	Acceptability of documentation to workers		
S4	Familiarity with documentation		
**External Factor**			
**Opportunities**		**Threats**	
O1	Traceability of infection through working procedures	T1	Increasing infectious disease risk and likelihood
O2	Accumulation of prior study cases via infectious disease epidemiology	T2	Depend on group capabilities for infectious disease management
**Decision Making Strategies**			
**Active response**		**Step-by-step implementation**	
SO1	Standardizing scenario	WO1	Continual revision of scenarios
SO2	Stipulation of scenario		
SO3	On-site application of stipulated scenario		
**Defensive response**		**Differentiation strategy**	
WT1	Minimize close/indirect contact through scenario		
WT2	Ensuring continuity of industrial roles of institution through scenario		

**Table 6 ijerph-19-12001-t006:** Scenario.

Part 1. General Requirement for Prevention of Infection and Transmission	
**1.1**	**Monitoring and record management**
(1)	Employees’ health status should always be monitored, and visible symptoms should be recorded and managed.
(2)	The issue of infectious diseases outside the organization is always monitored and considered.
**1.2**	**Risk source management**
(1)	Since the laboratory facilities have relatively high chance of indirect transmission, cross-contamination behavior and opportunities in workplaces should be avoided as much as possible.
(2)	Sufficient experimental utensils, tools, or equipment should be prepared as much as possible to control the possibility of fomite- or surface-mediated transmission.
**1.3**	**Exposure traceability**
(1)	To determine whether employees have direct or indirect contact with the primary case, procedures for reviewing an analytical testing manual, experiment or working diary, access record, etc. should be organized.
(2)	Infrastructure should be established to enable the implementation of the procedures proposed in this scenario.
**Part 2. Primary Case Occurrence**	
**2.1**	**Health quarantine, close observation, isolation**
(1)	In the case of primary case occurrence, it should be isolated in the workplace, assuming type of direct transmission (Type A), even before medical evaluation.
(2)	Close and indirect contacts should be closely observed until medical evaluation and (2.3) deduction. Since the discharge of agent takes place after the onset of symptoms, analyzing test activities is possible only if there are no similar symptoms.
**2.2**	**Record management**
(1)	If the primary case occurs, record symptoms and signs for further case occurrence situation. Propagation can be determined through record comparison even before the medical evaluation of primary case deduction.
(2)	Record management includes all symptoms of the body (fever, pain, vomit, food consumed, travel history) and backgrounds that can cause it.
**2.3**	**Medical evaluation and treatment**
(1)	The type of disease must be specified through medical evaluation and isolated from work until the results of the medical examination are derived.
(2)	Primary case should receive appropriate medical treatment.

## Data Availability

The data presented in this study are available on request from the corresponding author.

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
