# Peer review of "Worker Protection Scenarios for General Analytical Testing Facility under Several Infection Propagation Risks: Scoping Review, Epidemiological Model and ISO 31000"

_ijerph, 2022, doi:10.3390/ijerph191912001_

Round 1

Reviewer 1 Report

-The term industrial index negatively can be understood to refer to corporate profits (include examples)? Line 35.

-It talks about risks that may affect the normal operation of the company (lines 37, 40), it is very generic, I would like to quote some of them to make it clearer

-Its accreditation systems based on mutual 45 recognition agreements (MRA)

-Which characteristics of the environment (e.g. economic, social, technological, environmental...) and management systems does it refer to..., ) It is not known until section 2.1.2 that it refers to environmental characteristics. (line 73)

-The reference to Table 1.-3 is correct.  It seems to refer to Table 1.C which are sources of risks. (Line125)

-B) Confusing numbering (Line 144)

-It was difficult to state whether the use of sanitary gloves in laboratory procedures provides any advantage in suppressing transmission. (It is not explained why, which is unclear). (Lines 147-148)

-Table 3 at the end of paragraph line 221 does not interrupt the reading of the paragraph.

-Cite to which social fact or space of time it refers, I would cite example (Line 233)

-“Therefore, ‘risk sharing’ and ‘reducing likelihood (L) of occurrences’ also can be risk treatment option”. I think you are referring more to sharing time and workspace is what exposes you to risk, what you have to reduce is the time of online exposure. Clarify (Lines 249-250)

-(Table 5-Part 2) is right? Or  is it the table Table 6 Part 2. Check (Line 302)

-Missing final discussion

Author Response

Dear reviewer, Please see attached file

Reviewer 2 Report

Minor changes are suggested which can be found in attached document
